

# Rain Attenuation Prediction Model for Satellite Communications Based on The Météo-France Ensemble Prediction System PEARP

DAHMAN Isabelle[123], ARBOGAST Philippe[2], JEANNIN Nicolas[1], and BENAMMAR Bouchra[3]

[1]ONERA - DEMR, 2 Avenue Edouard Belin, 31055 Toulouse - France
[2]CNRM – GMAP, 42 avenue Gaspard Coriolis 31057 Toulouse - France
[3]CNES - DCT-RF-ITP, 18 Avenue Edouard Belin, 31400 Toulouse - France

**Correspondence:** Philippe ARBOGAST (philippe.arbogast@meteo.fr)

**Abstract.** This paper presents an example of usage of Ensemble Weather Forecast for the control of Satellite-based Communication Systems. Satellite communication systems become increasingly sensitive to weather conditions as their operating frequency is increasing to avoid electromagnetic spectrum congestion and enhance their capacity. In the microwave domain, electromagnetic waves that are conveying information are attenuated between the satellite and Earth terminals in presence of

hydrometeors (mostly rain drops and more marginally cloud droplets). To maintain a reasonable level of service availability, even with adverse weather conditions considering the scarcity of amplification power in spacecraft, fade mitigation techniques have been developed. The general idea behind those fade mitigation techniques is to re-route, change the characteristics, or re-schedule the transmission in case of too significant propagation impairments. For some systems, a scheduling on how to use those mechanisms some hours in advance is required, making assumptions on the future weather conditions affecting the

link. To this aim the use of weather forecast data to control the attenuation compensation mechanisms seems of particular interest to maximize the performances of the communication links and hence of the associated economic value. A model to forecast the attenuation on the link based on forecasted rainfall amounts from deterministic or ensemble weather forecasting is presented and validated. In a second phase, the model's application to a simplified telecommunication system allows demonstrating the valuable contribution of weather forecasting in the system's availability optimization or in the system's throughput

optimization. The benefit of using ensemble forecasts rather than deterministic ones is demonstrated as well.

## 1 Introduction and Background

Since a few decades, satellites have become absolutely essential in modern society. Their field of application is expanding constantly. Nowadays, they are widely used in various areas such as navigation, weather forecasting, disaster management, or telecommunications. In fact, geostationary telecommunication satellites can offer a global coverage which make them par-

ticularly attractive for bringing broadband Internet in isolated areas, where the access to terrestrial networks remains very limited.

     The data transmitted from satellites to Earth are conveyed by radiowaves whose frequency is comprised between 1 and 40 GHz. Frequencies within this range are classified into frequency bands, dedicated to specific applications for satellite





communications but also shared with other systems as fixed terrestrial wireless systems, microwave remote sensing instruments, radar or positioning systems. These frequency band labels (L,S,C,X, Ku,Ka and Q/V) are detailed in Table 1.

To increase the overall capacity of communication satellites (and hence the number of users of the system and/or the offered data rate), the use of the Ka and Q/V bands for which large modulation bandwidths are available is becoming widespread

among operational systems. However, the possibility to transmit data at a given data transmission rate is also dependent on the power level of the Electromagnetic wave received by the terminal. An insufficient power level will result in data losses. Power losses between the satellite and the Earth terminals are mostly caused by the dilution of the wave in space during its propagation and by some atmospheric phenomenons. In particular, atmospheric gases and more importantly the presence of hydrometeors attenuate electromagnetic waves. The scattering theories (Rayleigh or Mie) tell us that the level of the attenuation depends

on the ratio between the hydrometeor diameter and the wavelength (Gunn and East, 1954), (Oguchi, 1983)). This attenuation can reach several tens of decibels in case of liquid precipitations. Furthermore, for a given atmospheric state the attenuation tends to increase significantly with the frequency. The occurrence of propagation losses higher than a given threshold linked to the way the information is modulated results in data loss. To quantify this probability of data loss, the ITU (International Telecommunication Union) has established a climatology of the probability to exceed a given level of attenuation. The model

strives to quantify the probability of exceedance of tropospheric attenuation function of the radio-electrical characteristics of the link (frequency, elevation) and of the geographical position of the Earth terminal (ITU-R P.618-12, 2015). To this aim, climatological databases taking into account the rain regime (ITU-R P.837, 2012) are used. The height of the $0^{o}$C isotherm (ITU-R P.839, 2013) is also useful since solid hydrometeors have a negligible impact on electromagnetic wave propagation in the RF domain due to different electrical properties than liquid ones.

Figure 1 shows the attenuation due to rain exceeded 0.1 % of the time as computed by the ITU, for two different frequencies of transmission : 12 GHz and 50 GHz. The difference between the two scenarios is significant. At 12 GHz, the rain attenuation exceeded 0.1% of the time is a few dB for Europe whereas it is several tens of dB at 50 GHz. As the systems are usually designed to ensure a link availability larger than 99.9%, the attenuation values presented in Fig. 1 represent the power margins required to cope with the atmospheric impairments and to comply with the availability requirement function of the geographical

location of the receiving station.



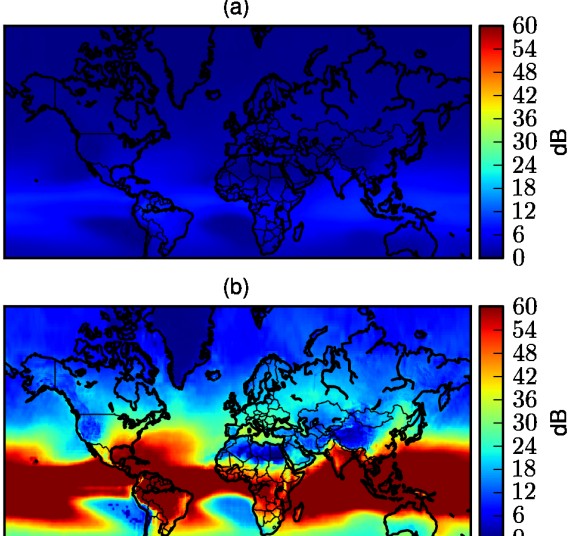

**Figure 1.** Attenuation exceeded 0.1% of the time computed from (ITU-R P.618-12, 2015) for a link with a geostationary satellite at 35 degrees of elevation at 12 GHz(a) and 50 GHz(b).

The margins required in some area to maintain the communication 99.9% of the time can reach 30 dB at Q/V band in mid-latitudes areas and even more in tropical areas. In other terms, it means that to ensure the availability of the link with a 99.9% probability a power 1000 times higher than the one required to maintain the link without rain is needed. As the power on-board of satellites is a scarce resource, techniques to mitigate adaptively the impairments have been developed (Panagopoulos
et al., 2004) to allow the efficient use of Ka and Q/V bands for satellite communications. These techniques rely either on signal re-routing (Jeannin et al., 2014) (i.e using another station ), delaying the transmission (Arapoglou et al., 2008), data rate decrease (Cioni et al., 2008), or in some extent in payload reconfiguration (Paraboni et al., 2009), (Resteghini et al., 2012). Some of them can be operated in closed loop. In this case the state of the channel is analyzed in real time and the decision to activate one or another mechanism is taken in near real-time. This is for instance the case of techniques using adaptive
coding and modulation that aims at adjusting the data rate and as a direct consequence the robustness of the communication link to the attenuation experienced by the communication link. One of the consequence is that a return communication channel between the Earth terminals and the satellite is required to convey the feedback. It is not necessarily available (for instance in broadcasting systems) and is also a cost and a security issue.

Other techniques require a forecast of the attenuation some hours in advance in order to prepare and to optimize the link
configuration through the telecommand of the satellite. In this respect the use of meteorological forecasts constitutes a promising approach to control the decision process associated with those fade compensation mechanisms. This has for instance been



studied in (Paraboni et al., 2009), (Biscarini et al., 2016) or (Jeannin et al., 2017) for different applications. One of the major difficulty highlighted by those studies is that the attenuation induced by rain on Earth space links is not strongly correlated to the model outputs due to the model accuracy and its limited space and time resolutions. Hence, it has been found that the use of the microwave attenuation computed directly from model outputs in the decision process shows poor skill.

To increase the attenuation prediction skill, probabilistic precipitation forecasts based on ensemble predictions could be used as long as cost-loss models are known and available.

The objective of this paper is to propose and describe a fully probabilistic approach to forecast rain attenuation by forecasting a probability of exceeding a given rain attenuation level rather than a deterministic value. To this aim Ensemble Prediction Systems (Descamps et al., 2015) will be shown to be particularly suited when using the probabilistic precipitation forecasts in

the control loop of a simplified communication satellite.

The organization of the paper is the following. The first section is devoted to the description of the model, where the different steps to obtain rain attenuation probability distributions conditioned to ensemble forecasts from Météo-France are described. In the second section, various scores are analyzed to assess the relevance of the proposed attenuation model. In a last section, the performances of the forecasts to maximize either the link capacity, the link availability or both are analyzed considering

concurrently measured attenuation data and the simulation of a simplified communication system.

## 2   DESCRIPTION OF THE ATTENUATION FORECAST MODEL

To develop the forecast model, it is needed to relate actual rain attenuation data to precipitation forecasts. The data used to this aim are detailed in a first part of this section. The elaboration and the development of the model is detailed in a second stage.

### 2.1   DATA

#### 2.1.1   Beacon Data

The attenuation due to rain on an Earth space-link can be characterized by measurement on Earth of power fluctuations of beacon signals (unmodulated signals) emitted by satellites. As the signal transmitted by the satellite has a constant power, the fluctuations of the received power are linked to the fluctuations of the tropospheric fade undergone by the signal during its propagation. Furthermore, the temporal scales of variation of water vapor, oxygen, clouds and rain attenuations differ

significantly which allows to discriminate the various contributors to the tropospheric attenuation. In particular, it can be deduced from the fluctuation of the beacon the rain attenuation affecting the transmission which dominates by large the total attenuation. Another possibility is to isolate the attenuation due to rain from the other components using concurrent radiometric measurements, that can be used to quantify clouds and gaseous attenuation.

ONERA conducts its own measurements campaigns analyzing the Ka band beacon signal of the Astra 3B geostationary

satellite in various experimental facilities. Throughout the years 2014 et 2015, Ka band attenuation measurements have been collected for two receiving sites located in Toulouse, France (latitude 43.5° N and longitude 1.5° E) (Boulanger et al., 2015)



and on Salon de Provence, France (latitude 43.6° N and longitude 5.1° E). These attenuation measurements are sampled at 1Hz but have been averaged on a 5-minute basis to save computational resources. An illustration of the Ka band rain attenuation time series used in the study is given in Fig. 2.

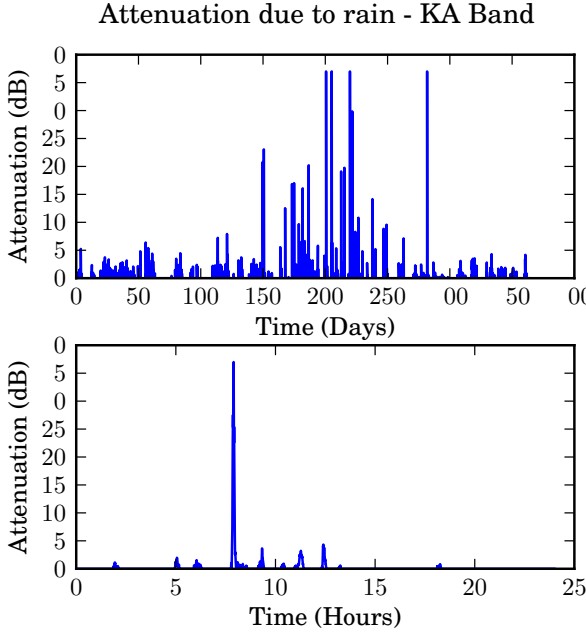

**Figure 2.** Ka band attenuation measurements operated by ONERA in 2014 ( upper Figure ) for a link between the geostationary satellite Astra 3B and a receiving station located in Toulouse, France. The lower figure focus on the 19 July 2014.

Those rain attenuation have been used as predictands in the statistical model discussed in part 2 as well as the probabilistic precipitation forecasts described in the next subsection.

### 2.1.2 Weather Forecast Model

The probabilistic precipitation predictions are built upon the sampling using the French global ensemble PEARP (Descamps et al., 2015). The horizontal resolution is variable and is about 10 km over France (60 km on the opposite side of the globe). Initial condition uncertainties are sampled using singular vectors and an Ensemble of 4DVar (Desroziers et al., 2014). The model error component is based on the multiphysic approach. For instance we run two different deep convection schemes (Bougeault, 1985). 35 members are calculated twice a day (at 0600 UTC and 1800 UTC). Both lagged runs are used together. Therefore, 70 members will be used as predictors in this study. Only 3 hourly precipitation forecasts available every 3 hours have been used on a 0.5°x0.5° grid. The ensemble forecast is denoted as $F$ in the following such as $F$=[member 1, member 2 .... member 70] where each member constitutes an estimation of the rain amount expressed in mm/3 hours.



## 2.2 The attenuation statistical prediction model

In order to provide a statistical link between predicted rain amount and rain attenuation, the members of the ensemble have been classified into the 5 categories, denoted $c_0$ to $c_4$ with $c_0 = [0., 0.01]$ mm/3h, $c_1 = [0.01, 0.1]$ mm/3h, $c_2 = [0.1, 1]$ mm/3h, $c_3 = [1, 6]$ mm/3h, $c_4 = [6, 50]$ mm/3h. Two areas of 100 km centered on the receiving stations located in Toulouse and Salon de Provence have been selected and the PEARP data were averaged on it. The temporal resolution of PEARP being of 3 hours, the attenuation forecasts thus obtained pertain as well to a 3 hours period. It should be stressed that the resolution of the Ensemble as well as the space and time resolution of the model output will not account for the time variability of the attenuation. In some cases the weather forecast is realistic but even a slight phase error may lead to the double-penalty problem (Nurmi, 2003). Therefore, we consider here a 100x100 km box around the attenuation observation site such as in (Theis et al., 2005).

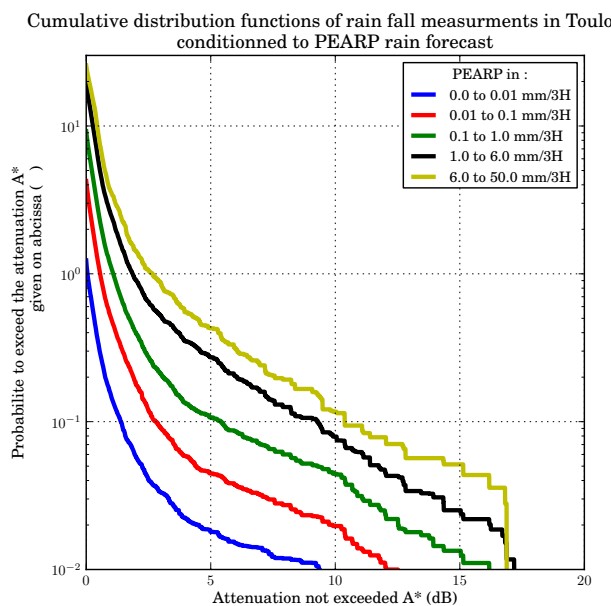

**Figure 3.** Probability to exceed the attenuation threshold given in abscissa based on data recorded in 2014 and 2015 in Toulouse, France.

By combining all members of the PEARP forecasts and the attenuation measurements of 2014 and 2015, the probability $P(A > A^*|F)$ of exceeding a given rain attenuation threshold $A^*$ knowing the precipitation forecast $F$, have been computed for Toulouse and is presented on Fig. 3. At each time horizon, the state of the channel is characterized by the 70 rain attenuation distributions corresponding to the 70 members available. A methodology must be defined in order to obtain a single probabilistic estimation of the rain attenuation occurring in future. The formula of total probability is written as follows:

$$P(A > A^*|F) = \sum_{i=0}^{n-1} P(F \in c_i) P(A > A^*|F \in c_i) \tag{1}$$





where A* is the attenuation threshold, expressed in dB. $F$ is the PEARP forecast with $F$= [member 1, member 2 .... member 70] . $c_0$ to $c_n$ are the PEARP classes previously described with $n$ being equal to 5. $P(A > A^*|F \in c_i)$ is the inverse cumulative distribution of attenuation, conditioned to the PEARP classes presented in Fig. 3. $P(F \in c_i)$ is the probability that, for a draw at a given time, the forecast belong to a given class. This probability can be directly sampled by counting the members of the

ensemble. . For example, let us assume that, for a given time horizon, half of the PEARP members belongs to the first PEARP class c0, half belongs to the second PEARP class $c_1$. In such case, $P(F \in c_0) = 0.5$, $P(F \in c_1) = 0.5$ and $P(F \in c_2, c_3, c_4) = 0$.

This methodology is equivalent to averaging the 70 rain attenuation distributions. An illustration is presented in Fig. 4. For the sake of simplicity, we consider only two realizations of the forecast. Both members lead to an inverse cumulative rain

attenuation distribution (black and green line) which, after averaging, leads to the probabilistic forecast of rain attenuation noted $P(A > A*|F)$ (gray line). Where A* is the attenuation threshold, in dB, exceeded and $F$ the PEARP forecast system. In this specific example F= [1.48,0.78] mm/3H.

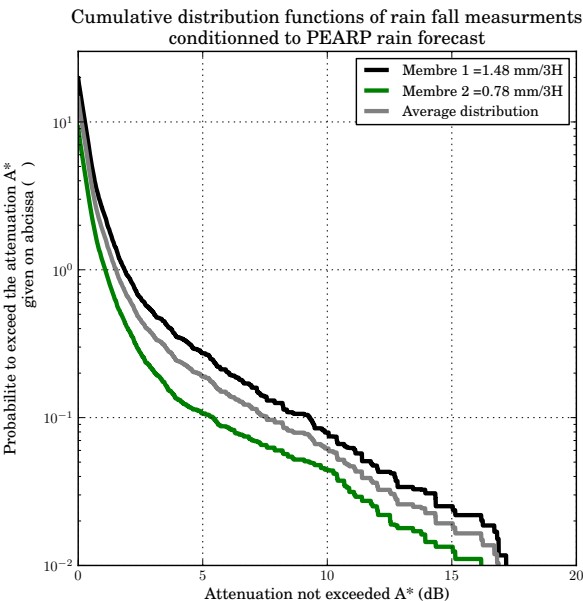

**Figure 4.** Probability to exceed the attenuation threshold given in abscissa. 2 distributions are selected according to the value of 2 ensemble members (black and green lines). The gray line stands for the average of those 2 distributions.

The use made of this predictive attenuation distribution will depend on the application of interest. One possible use is to get the attenuation threshold exceeded only 0.1% of the time, which is the tolerated unavailability threshold. This attenuation

threshold is equivalent to the power margin required to prevent inappropriate communication interruptions.





## 3 MODEL EVALUATION

The low spatial and temporal resolutions of the PEARP archives used for the learning process may cast some doubt on the utility of the attenuation forecasts. An appropriate use of the model requires an evaluation of its potential and weaknesses. A probabilistic forecast model is expected to present reliability and resolution. The reliability assesses the ability of a model to

provide a probabilistic prediction of a given event closed to the observed frequency of the same variable. The resolution is the ability to discriminate between events and non-events.

Only scores based on binary events are considered here. In a first stage, the reliability of the attenuation forecasts is addressed using the reliability diagram and the rank diagram also known as the Talagrand diagram. Resolution is evaluated using ROC curves and sharpness diagrams. The scores proposed here are computed using the available observations and model outputs

over the 2-year period defined above. Nevertheless, in order to evaluate the possible overfitting of the statistical model a bootstrapping approach is used: 6/10 of the sample is taken for the learning step and the remaining 4/10 for the score calculation. The procedure is repeated 100 times.

### 3.1 Model reliability assessment

The reliability of an ensemble model characterizes its ability to provide forecasts frequencies consistent with the observed

ones. For example, let's assume that the forecast system provides, for a particular event, a probability of occurring of $x\%$. Ideally, this event should be observed $x\%$ of the occasions on which such forecast is made.

The reliability diagram consists in plotting the observed frequencies against the forecast probabilities, previously classified into a few bins. For perfect reliability, the curve must merge with the diagonal line. A reliability curve located to the right of the diagonal line is typical of a model overestimating the probability of the event. Similarly, a model underestimating

systematically the probability event presents a curve located to the left of the diagonal line.

In the following, a positive event will be defined as the overrun of an attenuation threshold $A_{Th}$ alternatively set to 1 dB, 3 dB and 6 dB. The model developed provides the probability $P(A > A_{Th}|F)$ which is the predictive probability that a positive event occurs. Time series of $P(A > A_{Th}|F)$ are computed for the three values of $A_{Th}$ and for the two-year period. Figure 5 shows the corresponding reliability diagrams. The gray plain line indicates the perfect reliability.

It is as well common to represent the climatological probability which brings a complementary information on the model resolution. A forecast which provides the climatological probability has no ability for discriminating between cases of event and cases of non-events, this means that it has no resolution. The climatological probability is indicated by the dotted lines. Finally, the small windows show sharpness diagrams which are discussed later.





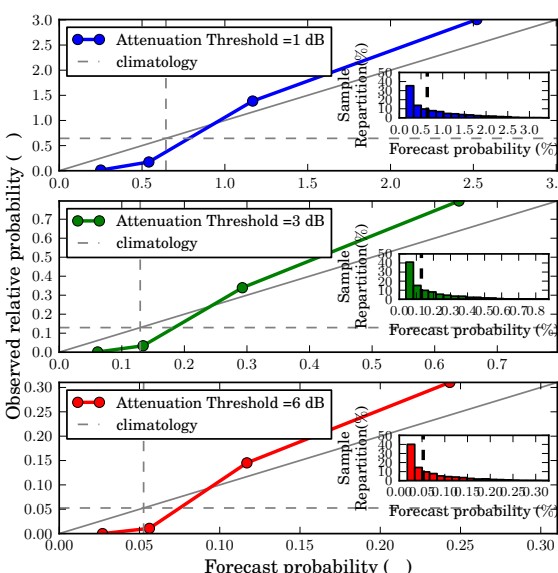

**Figure 5.** Reliability diagrams showing observed relative frequency as a function of forecast probabilities for attenuation thresholds 1dB (blue), 3dB (green) and 6 dB (red). Inset boxes indicate the frequencies of use of the forecasts. The dotted lines represent the climatological probability.

First of all, high probabilities are rarely met for those precipitation thresholds. That is why only low probabilities are shown. The shape of the reliability curves meets the expectations: the observed frequencies grow with the forecast probabilities and the curves deviate little from the diagonal line. This reflects the reliability of the attenuation forecast model and confirms its value for the forecast of the exceed of attenuation thresholds. However, the reliability of our statistical model is not perfect:

5  while low probabilities tend to be underestimated, high probabilities tend to be overestimated.

An other useful tool for determining the model reliability without considering thresholds is the rank diagram, also knows as the Talagrand diagram (Hamill, 2001). Since we have access to the cumulative distribution function discretized in $m$ quantiles rather than members we attribute the observation to its quantile. With regard to this, the abscissa of the rank diagram shown on Fig. 6 is as well discretized in $m$ bins. Non-zero values of precipitation (or attenuation) are often met for the few highest ranks

10  only. Then, considering our sampling this diagram could be considered as flat.





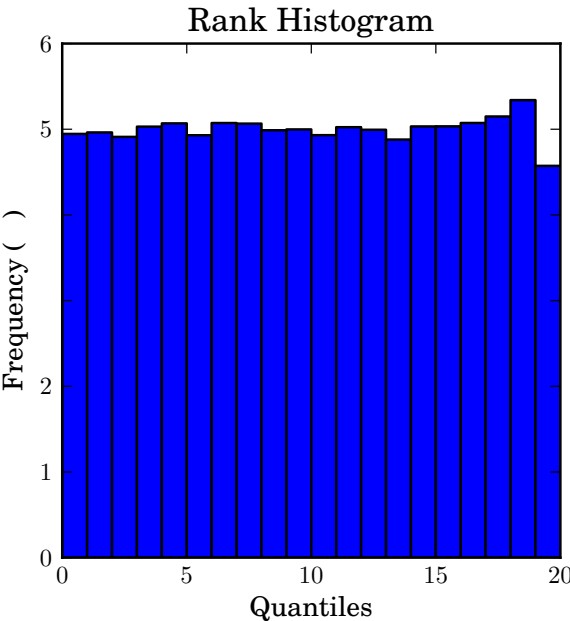

**Figure 6.** Rank histogram, also called Talagrand diagram, of probabilistic attenuation forecasts.

## 3.2 Resolution and sharpness

Resolution is another desired quality we expect from a probabilistic prediction. The Brier score decomposition tells us that resolution is the departure between the curves of Fig. 5 and the observation climatology represented by the horizontal dashed line. It turns out that the slope of our curves are steeper that the perfect-reliability curve suggesting pretty high resolution.

Sharpness is a property of the forecast alone. Nevertheless, it is also an indication of resolution since only sharp predictive distributions show high resolution. Here, the shape of the diagrams is consistent with a good sharpness although very low probabilities dominate.

Another approach to address the resolution as well as the value of a probabilistic prediction is to draw Receiver Operating Characteristic (ROC) curves. ROC graphs are widely used, in particular in the area of medicine (Zweig and Campbell, 1993),

to evaluate the power of discrimination of a classification model as a function of a decision threshold. These are of particular interest to evaluate and compare the performances of predictive models and are as well useful to determine the statistical decision-making thresholds (in terms of probability level) that will limit the impact of poor forecasting. As detailed in (Fawcett, 2006), the ROC analysis is based on the computation of the true positive and true negative rate for different classification thresholds. This method is applicable to binary classification problems only. The observation is either considered as belonging

to the positive or to the negative class. As a function of the decision-making threshold set, the classifier also assigns the prediction to one of these two classes. The confusion matrix between observations and forecasts presented in Table 2 lists the four outcomes: true and false positives, true and false negatives.





The true positive rate $TP_r$ represents the rate of true positive $TP$ among the total numbers of positive. It characterizes the sensitivity of the model, meaning its ability to predict an event when the event occurs.

$$TP_r = \frac{TP}{TP + FN} \tag{2}$$

The false positive rate, also called False Alarm rate $FA_r$, represents the rate of false positives $FP$ among the total number

of non occurring cases.

$$FA_r = \frac{FP}{FP + TN} \tag{3}$$

The ROC curves is a plot of $TP_r$ against $FA_r$ for a range of decision thresholds. The point whose coordinates are (0,0) is obtained for a classifier which systematically assigns the forecast to the negative class. Similarly, the point whose coordinates are (1,1) refers to a classifier which systematically assigns the forecast to the positive class. A trade-off must be considered

between the minimization of $FA_r$ and the maximization of $TP_r$. Ideally, without any knowledge about the cost/loss ratio, the ROC curve must approach the point (0,1) which allows an optimization of the ratio $TP_r/FA_r$ . On the contrary, a ROC curve approaching the diagonal from (0,0) to (1,1) indicates that the forecast skill is not better than chance level.

As described in part 2.2 the model evaluates $P(A > A_{Th}|F)$, the probability to exceed $A_{Th}$ as a function of weather forecasts. The problem is as follows: from which probability threshold $P(A > A_{Th}|F)$ the forecast should be considered as

positive?

The Fig. 7 illustrates the decision process involved in identifying positive forecasts as well as the possible scenarios leading to True Negative, False Negative, False Alarm and True Detection events.





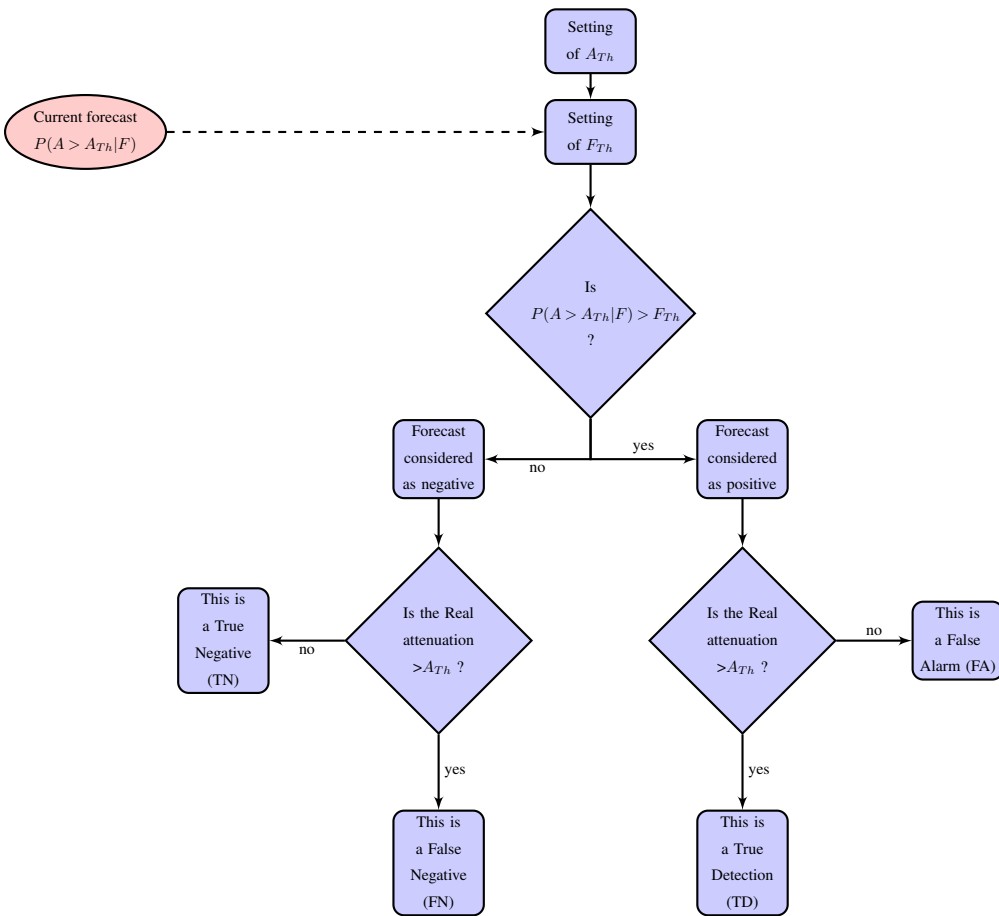

**Figure 7.** Block diagram of the discrimination process between positive and negatives forecasts.

Based on the process described in Fig. 7, false alarm and true detection rates have been computed for a range of $F_{Th}$, from 0.01 to 3 %, and for three attenuation thresholds $A_{Th}$ respectively set to 1, 3 and 6 dB. A 100-folds cross validation have been performed (the original sample was randomly partitioned into ten equal sized subsamples, six used as training data, 4 used as testing data). The average of the values computed in the loop leads to the solid lines in Fig. 8. The standard deviation of this

5   data is as well given by the 2D-box plot.



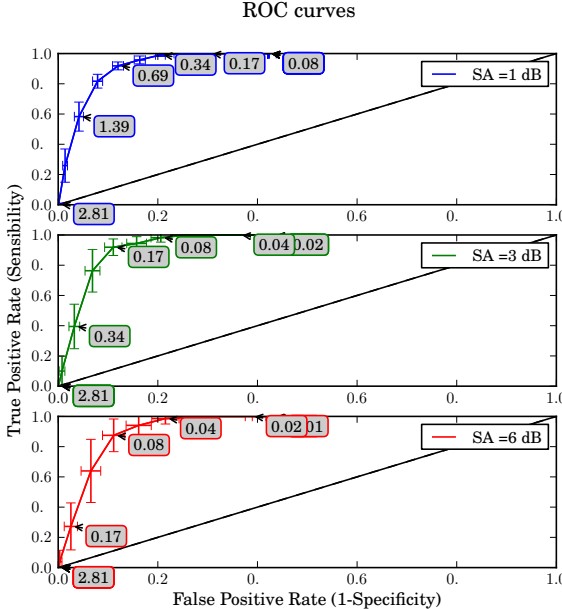

**Figure 8.** ROC curve obtained averaging 100-fold cross validation for attenuation thresholds set to 1dB(blue), 3dB(green) and 6dB(red). The box plots indicate the standard devision of each point. The diagonal line corresponds to random forecasts. A curves away from the diagonal represents a high levels of performance. Are indicated in the boxes, in percent, the forecast probability $F_{Th}$ associated to the points designated by an arrow.

The curves profile highlights the model ability to exploit the information provided by the weather forecasts. The first observation that can be made is that the ROC curves are located above the diagonal line (in black). This means that the model allows improvements over a strategy of randomly guessing the state of the forecast. The model is thus able to detect, in a rather accurate way, the over run of an attenuation threshold $A_{Th}$,whether is set to 1, 3, 6 dB.

5    It should be stressed that the departure from the diagonal of the ROC curve are equivalent for the three used values of $A_{Th}$. However, the standard deviation of the data appears to be highly dependent of $A_{Th}$. It turns out that the highest attenuation rate considered here is a rather rare event, the strong variability being related to the poor sampling of the ROC calculation.

Without any cost-loss ratio the potential value of the probabilistic binary forecast can be derived from the left and uppermost point of the curve. Actually, the cost-loss ratio is here non trivial and will be discussed in the section to come.

10  **4   MODEL APPLICATION**

In the previous part a methodology to develop a statistical forecast model of rain attenuation based on numerical weather forecasting has been detailed and evaluated. This final part is devoted to the description of methodologies for the optimization




of the offered capacity or of the economic value for a predetermined user oriented service offer. Both proposed methods use the attenuation forecast model outlined in section 2.

## 4.1 Optimization of radio resource management algorithms using weather forecasts

### 4.1.1 Link adaptation algorithms to propagation impairments

As detailed in 1, satellite transmissions become particularly sensitive to weather propagation conditions with growing frequencies. Especially in Ka and Q/V bands, attenuation due to rain may sometimes reach critical levels and cause the interruption of the communication. In such context, adaptive power control systems are particularly costly and can compensate only a limited portion of the tropospheric fades. It is however essential for satellite operators to ensure a minimal availability to the final user, typically set to 99.9% of the transmission time.

In order to maintain the link even in adverse propagation conditions without the need of radiating more power, an alternative is to adapt the link data rate to the weather conditions. The idea is to modify the Modulation and Coding used to carry the information as the tropospheric attenuation is varying. Here, the purpose is not to detail the Modulation and Forward Error Correction Coding techniques. More informations can be found in (Ziemer and Peterson, 2001) and (Watson et al., 2002). The whole point of this is that the nature of the modulation and coding combination used to transmit the information determines the

achievable capacity of the link. Furthermore, the use of a specific modulation and coding combination is subject to propagation conditions. In fact, the most efficient combinations are also the less robust to signal degradation. The challenge is therefore to select the most efficient coding and modulation suitable to maintain the link considering the experienced propagation conditions. Hereafter, the different combinations of modulation and coding available are referred to as MCS ( Modulation and Coding Scheme).

The characteristics of some MCS that are used in the following of the study are listed in the Table 3.The MCS are labeled from 1 to 6. The achievable capacity C is given as a function of the MCS selected for a bandwidth of 540 MHz.

The capacity characterizes the data amount, here in Gigabit, transmitted each second on the RF channel. An attenuation threshold A* indicating the maximal level of attenuation below which the MCS can be used to transmit the data stream without errors is also given. Those thresholds are of course depending on the characteristics of the link (bandwidth, wavelength) and

of the communication equipments (radiated power, antenna gains and receiver figure of merits).

To result in a valid transmission the attenuation threshold noted $A_{Th}$ associated to the MCS used must be higher than the experienced attenuation on the link. For example, the MCS n°6 should not be used unless the tropospheric attenuation is lower than 1 dB whereas the MCS n°1 can be used up to 10.8 dB of attenuation but the capacity of the link shrinks from 1.13 to 0.4 Gb/s. More generally, the most robust MCS may be used even while the tropospheric attenuation is strong and thus guarantees

the transmission in most of the experienced weather conditions. However, in that case the price of this resilience to adverse atmospheric conditions is a reduced capacity of the link. It is then preferable to adjust the MCS following weather conditions. For instance, an MCS offering high link capacities should be used under clear sky conditions when the tropospheric attenuation is weak.



It is understood that an inaccuracy in the estimation of the tropospheric attenuation leads to an inadequate selection of the MCS which can have significant consequences on the performances of the link. In case of overestimation of the tropospheric fading, a less efficient MCS that the one allowed by the real propagation conditions is selected. The difference between the capacity offered by the achievable MCS and the one used is lost. In case of the underestimation, the MCS selected does not allows to face the propagation impairments. Such a scenario inevitably leads to the interruption of the communication, namely to the unavailability of the link and a null capacity. Therefore, it is clear that the underestimation of the attenuation is far more prejudicial than an overestimation.

To illustrate this it can be assumed that the attenuation of the link reaches 2 dB. According to the Table 3, the MCS number 5 should be used to carry the information. In this scenario, the capacity of the link can reach 1 Gbit per second. Now, assuming that the attenuation has been overestimated using the weather forecast, the MCS n°4 would have been selected instead. The provided capacity would only reach 0.87 Gbit per second. This mistake would result in a capacity loss of 0.23 Gbit per second. On the contrary, assuming for the same experienced attenuation conditions an underestimation of the attenuation leading to the use of the MCS n°6 would lead the interruption of the communication and would results in a loss of 1 Gbit per second.

It is now clear that the efficiency and the availability of the satellite transmission is highly dependent of the MCS selection, based on the analysis of the propagation attenuation. Nonetheless, the propagation losses affecting the link are not always known as it requires a feedback. As a function of the available information on the state of the channel, different Modulation and Coding strategies can be applied. Theses are detailed in the next subsection.

### 4.1.2 Transmission strategies

The selection method of the MCS depends on the nature of the available information on the propagation channel. Four different scenarios have been considered with various assumptions on the type of information available to control the modulation and coding of the link:

- The propagation channel is perfectly known,

- The channel is unknown,

- Probabilistic weather forecasts are available,

- Deterministic weather forecasts are available.

The characteristic of those scenarios are summarized in the table 4.

In the first scenario, the attenuation experienced by the link is supposed to be known. The optimal strategy can be adopted, namely to dynamically select the most suitable MCS considering the current propagation conditions.

This strategy is referred to as ACM for Adaptive Coding and Modulation ((Zhu et al., 2006)). ACM technique is extremely efficient but requires a quasi-instantaneous feedback of the tropospheric attenuation affecting the channel to be sent to the modulator through a return channel. The existence of this return channel raises various issues in terms of payload complexity,





security of the telecommand (TC), and required infrastructure. In particular in the context data downlink of Low Earth Orbit satellite, it would need significant developments before entering in operation.

The second scenario assumes the total absence of information about the propagation conditions. When it is impossible to implement ACM, a constant coding and modulation (CCM) scheme is applied to the transmission. The objective is to use the MCS that will be compatible with the targeted availability. From (ITU-R P.618-12, 2015) the tropospheric attenuation not exceeded 99.9% of the time, noted $A_{99.9\%}$ can be computed. The CCM strategy consists in selecting the most efficient MCS satisfying the following condition : $A_{th} < A_{99.9\%}$. For example, let us consider a satellite system configuration operating in Ka band for which the tropospheric fading exceeds 8 dB, 0.1 % of the time. The CCM strategy would consists in applying as unique MCS, the numbered 2 in the Table 3. Only robust but relatively inefficient MCS are able in the context of Constant Coding and Modulation transmission to face the high tropospheric attenuations occurring in Ka and Q/V bands. An intermediate solution would be to schedule a plan of MCS depending on the probabilistic attenuation forecasts.

Third and fourth scenarii make use of the weather forecasts. The model developed and described in 2 gives the attenuation distributions conditioned to the PEARP forecasts. From those distributions, the corresponding attenuation thresholds exceeded 0.1% of the time can be deduced. It is proposed to choose the MCS in advance as a function of these probabilistic attenuation forecasts. In the following this strategy will be called PCM, for Programmed Coding and Modulation. Such strategy would provide a flexibility close to the one offered by an ACM link without requirement of a constant return link between the receiving stations and the satellite.Let us consider the specific case of Low Earth Orbit (LEO) Satellites. Once or several times a day, the LEO satellite comes in visibility during a few minutes of the receiving stations. None are able to transmit information to the satellite. However at pre-set times the satellite contact as well a control station equipped with a telecommand link which could be used to program a plan of MCS for the next orbit.

The Adaptive Coding and Modulation based scenario relying on currently experienced attenuation is obviously the most favorable. In theory, ACM strategy allows a perfect optimization of the capacity and highly limits the unavailability. Thus, the Programmed Coding and Modulation strategy based on the PEARP forecasts, is not expected to be as efficient as the ACM one. Here, the aim is rather to enhance the performances offered by a Constant Coding and Modulation strategy that relies only on local climatology without requirement of real attenuation measurements and of a return link. This point is discussed in the following subsection.

### 4.1.3 Results

As explained, the PCM strategy would be particularly suitable for the management of Low Earth Orbit Satellite transmissions for which return links are only implemented at the level of the control station, not at the level of the receiving stations. However, the lack of Ka band measurements for Low Earth Orbit Satellite prevents the model from being tested in this context. As a first step, an evaluation of the performances of the PCM strategy is thus proposed in the context of a geostationary satellite. A Ka band transmission line between the geostationary satellite Astra 3B and the receiving station located in Toulouse, France, have been simulated. The four scenarios lists in the table 3 have been tested.





The mission parameters used for the simulation are the following : a bandwidth of 540 MHz, an elevation of the satellite of 35° and a EIRP+G/T of 80 dBW/K. The EIRP, for Effective Isotropic Radiated Power, measures the ability of the transmitted antenna to direct the power emitted in a given area. The term G/T is a figure of merit in the characterization of ground station antenna performance, where G is the antenna gain in decibels at the received frequency, and T is the equivalent noise

temperature of the receiving system in kelvins.

The PEARP forecasts of 2014 and 2015 have been used as input of the PCM decision algorithm. The ACM strategy have been based on the Ka band measurements of the same years assuming an idealized adaptation of the MCS to the channel state.

The Fig. 9 shows the mean capacities offered by the ACM, CCM and PCM strategies. In order to assess the interest of using probabilistic forecasts over deterministic ones, the PCM strategy has also been tested by using the control PEARP member

only.

The target availability has been set to 99.9% of the communication time. Here again a bootstrapping approach has been used. The available samples were partitioning in 10 subsets, 6 constituting the training dataset, 4 constituting the testing set. The procedure has been repeated 100 times.

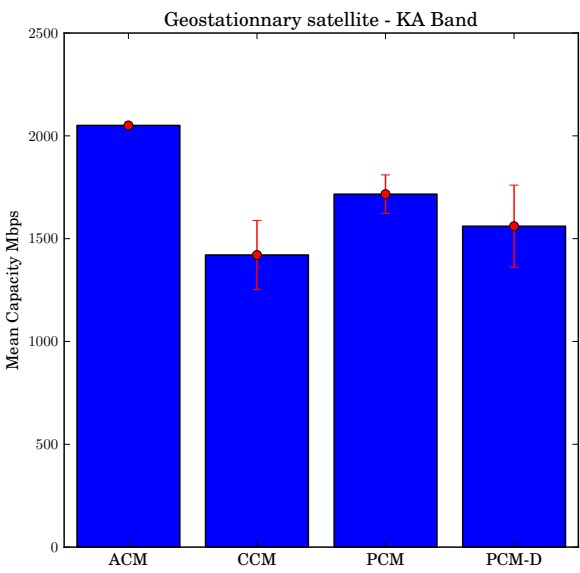

**Figure 9.** Mean capacities obtained for a target availability of 99.9% and considering a GEO satellite. Comparison of ACM, CCM, PCM and PCDM deterministic strategy. The box plots indicate the standard devision of the data.

As expected, the best performance have been obtained for the ACM strategy which is an upper bound. Due to limited weather

forecast predictability, the proposed PCM strategy is not able to offer the same level of capacity. Nevertheless, it is clear that,




with an increase of the capacity close to 17 %, this technique significantly improves the throughput ensured by a constant coding and modulation without any prior on the instantaneous state of the propagation channel.

It also appears that the use of ensemble forecast outperforms the use of deterministic forecast. The results obtained are highly dependent on the systems parameters such as the satellite elevation, the ground station location or the transmission frequency.

Furthermore, the use of higher frequencies (Q/V bands) or of receiving station located in tropical areas would inevitably result in even more noticeable differences in the throughputs achievable.

So far, the methodology proposed, consisting in programming in advance the modulation and coding using weather forecasts, requires to set a level of target availability. In fact, the MCS are choose in order to ensure this availability. Without this constraint the number of perspectives would certainly be increased. In the next part, a more general approach is adopted. A decision

process based on the optimization of an economic value taking into account both capacity and availability offered is proposed.

### 4.1.4 Economic value of the forecast

When it comes to use a probabilistic forecast, the decision to change or not the link MCS amounts to search the forecasted probability above which the forecast is considered as positive. In this context the issue is to find the forecast probability $P(A > A_{Th}|F)$ above which it is considered that the attenuation threshold $A_{Th}$ is exceeded, and consequently above which

an action must be taken.

So far, the required availability of the link has been set to 99.9% in order to respond to the operator's needs. In such context, the probabilistic attenuation forecast has to be considered as positive when the chance to exceed the attenuation threshold of interest $A_{Th}$ are equal or higher than 0.1 %, which is the value of the tolerated unavailability. In fact, the choice of a decision threshold higher than 0.1% would not allows to bring the required availability. On the contrary, a lower decision threshold

would result in a capacity loss.

This requirement of 99.9 % of availability is typical for communication satellites, since the tolerance of the final users to communication outages is highly limited. However, it is easy to imagine further applications for which both availability and mean capacity have to be optimized without prerequisites on any of these parameters. The challenge in this case is to determine the optimal decision threshold, noted $D_{Th}$ in the following.

As an example, still considering the particular case of an Earth observation satellite. The images acquired by the satellite on its path must be transmitted on Earth as soon as a receiving station is in visibility. The visibility periods of the Earth stations are limited to a few minutes. Especially under rainy conditions, it could happen that the data sent by the satellite do not reach the receiving station. In this case data are definitively lost. It might then be sometimes more careful to temporary store the data in the on-board memory, waiting for the next contact with the ground. This strategy has been evaluated in (Jeannin and

Dahman, 2016). In this context the probabilistic attenuation forecasts could be of great value to decide in advance either the data transmission or the data storage, provided that the decision threshold is chosen carefully.

A methodology to determine $D_{Th}$, defined here as the forecast probability from which the data storage is preferred to the data transmission, is now proposed. The optimal value of $D_{Th}$ is the one that allows both, the maximization of the amount of transmitted data and the minimization of the amount of lost data. For the sake of simplicity, it is considered that the data sent



can only be received if the attenuation do not exceed a given attenuation threshold $D_{Th}$. A perfect decision algorithm would result in the data transmission when the real attenuation is lower than $A_{Th}$ and otherwise in the data storage.

The ROC curves introduced in part 3 could be used to determine the probability forecasts threshold that minimize the false alarms rate while maximizing the true detection rate (Greiner et al., 2000). Such method shows its limitations once it is required
5   to penalize either False Negative or False positive events. We rather propose here a decision process based on the optimization of an economic value that one could adapt as a function of the needs of the systems targeted.

To account for the high cost of lost data and for the successfully transmitted ones, the economic value to be maximized, noted EV is defined in Eq. (4).

$$EV = C \cdot 10^{-L} \tag{4}$$

10   Where C is the total amount of transmitted data and L is the fraction of lost data.

Decreasing the value of $D_{Th}$ limits the amount of transmitted data as well as the amount of lost data and thus impacts the Economic Value. In the Fig. 10 is illustrated the evolution of EV as a function of $D_{Th}$ for several attenuation thresholds $A_{Th}$.

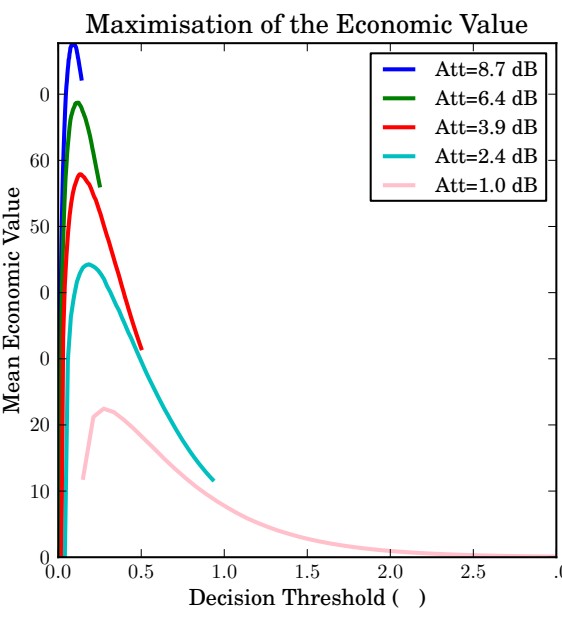

**Figure 10.** Evolution of the economic value defined in equation 4 as a function of the decision threshold used to discriminate between positive and negative forecasts.

It appears in the Fig. 10 that, for all attenuation threshold, an optimal threshold $D\_Th$ can be found. It should be used for deciding either the data transmission or the data storage. As example, for an attenuation threshold of 1.0 dB, the optimal





decision threshold is 0.24 %. This means that when the tropospheric attenuation is tolerated up to 1.0 dB, the appropriate strategy would consist in transmitting data only when the probability $P(A > 1dB|F)$ is below 0.24% . Otherwise the data should be stored. The ROC curve in Fig. 8 tells us that this decision threshold corresponds to a fair amount of false alarm. This can be related to the large cost of the misdetection (lost data $L$) set in the definition of the economical value (Eq. (4)).

## 5  CONCLUSION AND PERSPECTIVES

This study has presented a methodology for predicting the rain attenuation which affects the satellite transmissions. The sensitivity of satellite transmission to rain becomes particularly sensitive with the on-going trend to use high frequency bands, from 20 to 50 GHz. The proposed model exploits the probabilistic rain forecasts of Météo-France short-range ensemble prediction system PEARP and delivers probabilistic attenuation forecasts at 20 GHz. In particular due to the inhomogeneity, in terms of temporal resolution, of the predictand used for the model's learning process, a bias into the model was expected. It turns out that reliability diagrams show forecasts frequencies close to the observed ones. The figures obtained suggest that the statistical model shows only a small remaining bias. For a more complete assessment, ranks diagrams and ROC curves demonstrate the model ability to discriminate between event and non event cases and to give forecasts frequencies different from the climatology one. Consequently, it can be concluded that the model shows satisfactory reliability resolution and sharpness.

In satellite communication, the main concerns are the link availability and capacity. The primary hypothesis tested in the study was that the probabilistic weather forecasts could be very helpful to maintain the high availability required by the satellite operators while optimizing as far as possible the capacity. It has been shown that to condition the type of waveform (modulation and coding scheme) used to transmit the information, to probabilistic weather forecasts allows increasing the mean capacity of the link while ensuring the availability of 99.9 % usually required. It has also be proven that the benefit is higher using probabilistic weather forecasts over deterministic one.

The link availability and capacity are highly interdependent. Within a certain limit, increasing one of these parameters is detrimental to the other one. The request of a high availability inevitably results in a limitation of the capacity which may be in some context particularly unfortunate. It would then be sometime profitable to be able to find the least expensive combination of these two parameters.

In a last stage, a strategy to maximize an economic value accounting for the transmitted data volume and for the fraction of successfully transmitted data has been proposed. This economic value could be adapted to the targeted application. For this initial attempt to optimize high frequency band satellite transmissions from ensemble weather forecast systems, encouraging results have been obtained. It should be stressed that the application to the forecast of rain attenuation around 50 GHz, or to more sensitive ground station locations such as in tropical regions could show more value. Unfortunately, we do not have any attenuation observations for those contexts.

It has also to be mentioned that the horizontal resolution and the temporal resolution of the PEARP forecast as well are a non-negligible drawbacks. It would make sense to replace the global ensemble PEARP by the regional mesoscale ensemble prediction system PEARP (Bouttier et al., 2011) based on AROME in the attenuation prediction process. The authors would





like to stress the fact that the time resolution of the model output used for the prediction process should be higher than 3 hours are probably even higher than 1 hour according to the large time-variability of attenuation observations.

As a conclusion, though perfectible, the model developed allows to demonstrate the benefit of using ensemble weather forecasts in the field of satellite communications. The wide range of applications of the model developed includes the following, particularly relevant weather dependent applications which could be address in further publication:

- site diversity (Jeannin et al., 2014) for which an anticipation of the attenuation would allows to schedule in advance the switching from the ground station under unfavorable propagation conditions to redundant ones.

- deep space links (Biscarini et al., 2016) for which an anticipation of the attenuation would allow scheduling the data transmission when the propagation conditions are the most favorable.



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




| Band | Frequency range | Applications for satellite communication systems | Bandwidth available for communication satellites |
|------|-----------------|--------------------------------------------------|--------------------------------------------------|
| L | 1-2 GHz | Mobile Satellite services | Some tens of MHz |
| S | 2-3 GHz | Mobile Satellite services | Some tens of MHz |
| C | 4-6 GHz | TV Broadcasting and point to point communication link | $\approx$ 200 MHz |
| X | 7-9 GHZ | Military communication systems | 1 GHz |
| Ku | 11-16 GHz | TV Broadcasing, point to point link, internet access | 1.5 GHz |
| Ka | 20-30 GHz | Internet access | 2.5 GHz |
| Q/V | 40-50 GHz | Future systems for Internet access | 5 GHz |

**Table 1.** Frequency bands used for satellite communications.





|  |  | Observation | |
| --- | --- | --- | --- |
|  |  | Positive | Negative |
| Forecast | Positive | **True Positives** ($TP$) | **False Positives** ($FP$) |
|  | Negative | **False Negatives** ($FN$) | **True Negatives** ($TN$) |

**Table 2.** Confusion Matrix for ROC curves construction





| Number of the MCS | Modulation | Code Rate | Attenuation threshold $A_Th$ (dB) | Capacity achievable $C$ (Gb/s) |
|---|---|---|---|---|
| 1 | Q-PSK | 0.36 | 10.8 | 0.4 |
| 2 | Q-PSK | 0.51 | 8.7 | 0.55 |
| 3 | 8-PSK | 0.47 | 6.4 | 0.71 |
| 4 | 8-PSK | 0.62 | 3.9 | 0.87 |
| 5 | 8-PSK | 0.70 | 2.4 | 1 |
| 6 | 16-APSK | 0.79 | 1 | 1.13 |

**Table 3.** Extract of possible modulation and coding schemes as defined in the Digital Video Broadcasting - Satellite - Second Generation (DVB-S2X) standard (DVB, 2014) and of their operating characteristics.



| | Scenario n°1 : **The real attenuation is known** (1 data every 5mins) | Scenario n°2 : **State of the channel unknown** | Scenario n°3 : **Only PEARP forecasts available** (1 data every 3 hours) | Scenario n°4 : **Only Deterministic forecasts available** (1 data every 3 hours) |
|---|---|---|---|---|
| Coding and Modulation Strategy | **ACM** Adaptive Coding and Modulation | **CCM** Constant Coding and Modulation | **PCM** Programed Coding and Modulation based on PEARP forecasts | **PCM-D** Programed Coding and Modulation based on deterministic forecasts |
| Model required | None | Climatology of Ka band attenuation (ITU-R P.618-12, 2015) | Ka band attenuation forecast model based on PEARP forecasts | Ka band attenuation forecast based on deterministic forecasts |
| MCS Decision Factor | **Real Attenuation** | **Statistical Attenuation threshold** | **Statistical Attenuation threshold** conditioned to PEARP forecast | **Statistical Attenuation threshold** conditioned to the deterministic forecast |
| Decision time | **Continuously** | **Once** When the satellite is lunch | **At days -1 or days -2** | **At days -1 or days -2** |
| Availability ensured | **Close to 100.0%** | **Close to 99.9%** | **Close to 99.9%** | **Close to 99.9%** |

**Table 4.** Proposition of Coding and Modulation strategies as a function of the available information on the state of the channel.

*Competing interests.*