# Peer review of "Rain Attenuation Prediction Model for Satellite Communications Based on The Météo-France Ensemble Prediction System PEARP"

_Natural Hazards and Earth System Sciences, 2018_

## Referee Comment (RC1) · Anonymous Referee #1 · 11 Jul 2018

The paper "Rain Attenuation Prediction Model for Satellite Communications Based on The Météo-France Ensemble Prediction System PEARP" by Dahman and co-Authors presents a study on the use of a Ensamble weather modeling system to manage and control Satellite Communication systems. The Author make use of the precipitation fields produced by PEARP to estimate and forecast possible attenuation of the satellite signal, with the aim to optimize microwawe based data transfer, with relevant economic impact.

The paper is surely interesting and well writtem, with well suppoorted conclusions. However, I think that its subject is rather far from the aim and scope of this journal. No

reference to natural hazards or risks is reported: the subject of the paper seems to be an innovative application of meteorological forecast products. I suggest to resubmit the manuscript to another Journal more focused on weather forecast application, or, to better contextualize the work in a hazard framework, if they still want to resublmit to NHESS. For this reason, I sugget the rejection of the manuscript in the present form.
* * *

---

## Referee Comment (RC2) · Anonymous Referee #2 · 13 Jul 2018

The paper presents a model to compute rain attenuation statistics from ensemble weather forecasting that can be used to optimize the design of satellite-based communication systems. The statistics are obtained combining the members of the PEARP ensemble prediction system and the attenuation measurements from the Ka band beacon signal of the Astra 3B geostationary satellite for the years 2014 and 2015. The model reliability is evaluated with the Astra 3B measurements in terms of forecasted and observed probability. The model resolution is investigated by resorting to the Receiver Operating Characteristic curves. The advantages of using ensemble weather forecast with respect to the classical approach based on climatological statistics (i.e., from ITU-R) are shown.

[Figure]

GENERAL COMMENTS: The paper is interesting, well written and well organized. My judgement is positive. However, some general minor corrections are needed and some details about the procedure are missing or not clear (see specific comments).

Please also note the supplement to this comment:
https://www.nat-hazards-earth-syst-sci-discuss.net/nhess-2018-94/nhess-2018-94-RC2-supplement.pdf

**Supplement:**

**Rain Attenuation Prediction Model for Satellite Communications Based on The Météo-France Ensemble Prediction System PEARP**

DAHMAN Isabelle [1,2,3], ARBOGAST Philippe [2], JEANNIN Nicolas [1], and BENAMMAR Bouchra [3],

[1] ONERA - DEMR, 2 Avenue Edouard Belin, 31055 Toulouse - France

[2] CNRM – GMAP, 42 avenue Gaspard Coriolis 31057 Toulouse - France

[3] CNES - DCT-RF-ITP, 18 Avenue Edouard Belin, 31400 Toulouse - France

The paper presents a model to compute rain attenuation statistics from ensemble weather forecasting that can be used to optimize the design of satellite-based communication systems. The statistics are obtained combining the members of the PEARP ensemble prediction system and the attenuation measurements from the Ka band beacon signal of the Astra 3B geostationary satellite for the years 2014 and 2015. The model reliability is evaluated with the Astra 3B measurements in terms of forecasted and observed probability. The model resolution is investigated by resorting to the Receiver Operating Characteristic curves. The advantages of using ensemble weather forecast with respect to the classical approach based on climatological statistics (i.e., from ITU-R) are shown.

General comments

The paper is interesting, well written and well organized. My judgement is positive. However, some general minor corrections are needed and some details about the procedure are missing or not clear (see specific comments).

Specific comments

- Pag.2 (Introduction): note that the ITU-R recommendations that are cited must be updated with ITU-R 618-13, 2017 and 837-7, 2017.
- As concerning the recommendation ITU-R 618, which is further cited and adopted in the work, please check that the results are in line with the last recommendation 618-13, 2017 (e.g., the CCM results presented in section 4.1.3).
- Par. 2.1.1 (Beacon data): which is the precise frequency of Ka band signal of the Astra 3B satellite?
- Par. 2.1.2 (Weather forecast model), please add some additional details concerning the adopted PEARP system:
    - Please specify which is the total period over which the PEARP model is run: it should be over the years 2014 and 2015 (where the beacon measurements are available) but this detail should be explicated in this paragraph to give a complete presentation of the set-up of the adopted weather forecast model.
    - The lead time of the weather forecast is not specified: since the ensemble forecast F includes 70 members per day (35 computed at 06.00 UTC and 35 computed at 18.00 UTC), I guess that the we are dealing with daily weather forecast (i.e., 70 members per each day of the forecasted period) but it should be better clarified.
    - It is not clear if the PEARP members are time-series, over a certain period, of the rain accumulated every 3 hours: please, clarify this point.
    - Please write explicitly in this paragraph that the time resolution of the weather forecast is 3 hours (which is the availability of the forecasted cumulated rain).
    - Line 11, "Both lagged runs are used together": please clarify this sentence.
- Par. 2.2 (The attenuation statistical prediction model):
    - Pag.7, line 2: it should be called "complementary cumulative distribution of attenuation" instead of "inverse cumulative distribution of attenuation".
    - How is it computed the complementary cumulative distribution of attenuation conditioned to the PEARP classes (the right probability of eq.1)? The probability that A>A* (from the beacon measurements) is combined with the condition on PEARP classes but the procedure is not clear: measurements averaged over 5-minutes are compared with model outputs available every 3-hours.

- o Pag.7, line 8, "This methodology is equivalent to averaging the 70 rain attenuation distributions": this is true for a certain time horizon. It is not clear if the equation (1) is computed per each day of the simulated period (2014-2015).
  - o Concerning this section 2.2 and the applicability of the equation (1), some clarifications should be done. If I understood right, the available beacon measurements, combined with the forecasts (computed within the same period of measurements), are used to compute the probability $P(A>A^*|F \in c_i)$. Once this probability is computed, it is stored such a "library" available in the operative context. When the satellite communication must be designed, a new forecast is produced for the target satellite-to-Earth transmission period. This forecast is used to compute $P(F \in c_i)$ that, together with the probability in the library, allows the computation of the total probability in eq. (1). Please add this details in the paper.
- Par. 3.1 (Model reliability assessment):
  - o Pag. 8 line 23, "Time series of $P(A>A_{th}|F)$ are computed": please clarify how are obtained the time series of the probability (and how are computed the curves of fig. 5). Once fixed $A_{th}$ it comes that $P(A>A_{th}|F)$, for a given F, is a number.
  - o Fig.5: why is the climatological probability represented by a horizontal and a vertical line?
  - o Fig.6: please clarify which is the rationale of the rank diagram.
- Par. 3.2 (Resolution and sharpness):
  - o All the symbols (TP, FN, FP, TN, FA) must be introduced and defined before being used.
  - o Fig.7: please check the block diagram, I guess that the (TN) and (FN) boxes should be inverted. Please check the consistency between the symbols used in diagram and the ones used in the equations (2) and (3): TP should be used instead of TD.
  - o Pag.12: Please give a definition of $F_{th}$ and explain how is it chosen.3
  - o Pag.13, lines 8-9: please clarify the sentence.
- Par. 4.1.2 (Transmission strategies): the deterministic scenario is not described.
- Table 4: some details should be given regarding the decision time, especially for the scenarios 3 and 4.
- Par.4.1.3 (Results):
  - o How was computed the "Mean" capacity of the different scenarios?
  - o Pag.17, lines 8-10: please add some details.
- Par.4.1.4 (Economic value of the forecast):
  - o It is not clear the meaning of $D_{th}$: in line 32 (pag.18) is defined as a threshold on the forecast probability but in line 1 (pag.19) it seems to be a threshold on the attenuation.
  - o Pag.19, line 2: is it $A_{th}$ or $D_{th}$?
  - o Pag.19, eq. (4): please add a reference for the equation (4) and clarify L (I guess it is total lost data over total transmitted data).
  - o Fig.10: why is the y axis a "mean" value? Is it averaged over the 2 years (2014-2015) of simulations and measurements?
  - o Pag.20, line 4: please explain better.

Technical corrections

- Pag.2 (Introduction): please add a reference for the frequency band division in Table I.
- Pag.4, par.2.1.1, lines 25-27: the sentences are not very clear, please rephrase.
- Pag.8, line 8: the acronym ROC is introduced here for the first time but it is not defined.
- Pag.10, line 4: please check typo error.
- Pag.11, line 11-12: the sentences is not clear, please rephrase.
- Pag.14, line 23: probably the symbol $A^*$ should be replaced by $A_{Th}$.
- Pag.16, line 33: check typo errors and change table 3 with table 4.
- Pag.17, line 2: "transmission" instead of "transmitted"
- Fig.9: please check typo error in the caption
- Pag.21, lines 1-2: please check typo errors

---

## Author Comment (AC1) · 12 Sep 2018

We sincerely thank you for the constructive criticisms and valuable comments, which will be of great help in revising the manuscript. Please find below the detailed reply to the comments in https://www.nat-hazards-earth-syst-sci-discuss.net/nhess-2018-94/nhess-2018-94-RC2-supplement.pdf .

- Pag.2 (Introduction): note that the ITU-R recommendations that are cited must be updated with ITU-R 618-13, 2017 and 837-7, 2017. As concerning the recommendation ITU-R 618, which is further cited and adopted in the work, please check that the results are in line with the last recommendation 618-13, 2017 (e.g., the CCM results

presented in section 4.1.3).

References to ITU recommendations will be updated.

- Par. 2.1.1 (Beacon data): which is the precise frequency of Ka band signal of the Astra 3B satellite?

The Ka band attenuation observations used in the paper measure the 20 GHz beacon of the Astra 3B Satellite.

-Please specify which is the total period over which the PEARP model is run: it should be over the years 2014 and 2015 (where the beacon measurements are available) but this detail should be explicated in this paragraph to give a complete presentation of the set-up of the adopted weather forecast model. o The lead time of the weather forecast is not specified: since the ensemble forecast F includes 70 members per day (35 computed at 06.00 UTC and 35 computed at 18.00 UTC), I guess that the we are dealing with daily weather forecast (i.e., 70 members per each day of the forecasted period) but it should be better clarified. o It is not clear if the PEARP members are time-series, over a certain period, of the rain accumulated every 3 hours: please, clarify this point. o Please write explicitly in this paragraph that the time resolution of the weather forecast is 3 hours (which is the availability of the forecasted cumulated rain). o Line 11, "Both lagged runs are used together": please clarify this sentence.

The complementary distribution function of attenuation conditioned to the PEARP forecasts is computed from the daily PEARP forecasts archived in 2014 and 2015. We use together 36h forecast run at 1800UTC day D together with 24h forecasts run at 0600 UTC D+1. The 35 members of both runs, respectively 06.00 UTC and 18.00 UTC, have been used indiscriminately (we say that runs are used together). Only three hours cumulative rain rate forecasted are available and are used to predict attenuation a finer time resolution. . .. These points will be clarified in the paper.

-How is it computed the complementary cumulative distribution of attenuation conditioned to the PEARP classes (the right probability of eq.1)? The probability that A>A* (from the beacon measurements) is combined with the condition on PEARP classes but the procedure is not clear: measurements averaged over 5- minutes are compared with model outputs available every 3-hours.

The temporal dynamics of attenuation due to rain is high. It would not make sense to average rain attenuation data on a 3-hour basis. Furthermore, this would dramatically reduce the size of the training dataset used to compute the complementary distribution function of attenuation conditioned to the PEARP forecasts. The strategy adopted here consists of building a contingency table between the rain attenuation measurements available every 5 minutes and the PEARP forecasts available every 3 hours. For that, each PEARP forecast is duplicated 36 times (because there is 36 times 5 minutes in three hours). The contingency table allows computing the probability of exceeding, in average during 5 minutes, a given attenuation threshold knowing the PEARP forecasts computed on the same period of reference. The present approach of statistical calibration could be understood as a space AND time downscaling as well.

-Pag.7, line 8, "This methodology is equivalent to averaging the 70 rain attenuation distributions": this is true for a certain time horizon. It is not clear if the equation (1) is computed per each day of the simulated period (2014-2015). In part 2, a methodology to compute the probability of exceeding a given attenuation threshold from a unique value of predicting rain amount is proposed. Nonetheless, the PEARP forecasts are composed of 70 members (35 from each run). In an operational context, the translation of PEARP forecasts into attenuation forecasts will then result in 70-attenuation distributions. Equation 1 consist in averaging theses 70 attenuation distribution, along the probability axis, in order to obtain a unique predictive attenuation distribution.

-Concerning this section 2.2 and the applicability of the equation (1), some clarifications should be done. If I understood right, the available beacon measurements, combined with the forecasts (computed within the same period of measurements), are used to compute the probability $P(A>A^*|F$ âĹĹ $c_i)$. Once this probability is computed, it is stored

such a "library" available in the operative context. When the satellite communication must be designed, a new forecast is produced for the target satellite-to-Earth transmission period. This forecast is used to compute P(F âĹĹ ci) that, together with the probability in the library, allows the computation of the total probability in eq. (1).

Thank you for the above reformulation which is perfectly correct and will improve the paper.

-Fig.5: why do a horizontal and a vertical line represent the climatological probability?

The reliability diagram shows the observed frequencies of an event as a function of its forecast probability. The diagonal line indicates the perfect reliability. It is conventional to represent the climatological probabilities of the event in the forecasts by the vertical line and in the observations by the horizontal line. -Please clarify which is the rationale of the rank diagram.

In the case of an ensemble prediction system, the rank histogram of the position of the verifying observation with respect to the predicted ensemble values provides a measure of reliability (Talagrand et al. 1999). When the ensemble is reliable, say the verification and the evaluated system are from the same distribution the diagram is flat. If the diagram follows a U-like shape then the observation is often outside the ensemble suggesting the ensemble is underdispersive. . .

-Fig.7: please check the block diagram, I guess that the (TN) and (FN) boxes should be inverted. Please check the consistency between the symbols used in diagram and the ones used in the equations (2) and (3): TP should be used instead of TD.

Thank you for pointing that out. This will be corrected. - Pag.12: Please give a definition of Fth and explain how is it chosen. o Pag.13, lines 8-9: please clarify the sentence.

Fth the is threshold from which the predictive probability of exceeding a given attenuation threshold is considered significant enough for establishing protectives measures (for example to reduce the link capacity). In the simplest case, it is the mean when the

costs of False Alarms are in the same order of magnitude of the non-detection's one. Fth can be deduce from the ROC curves. The optimal Fth is then the one given by the left and uppermost point of the curve (each point of the ROC curve correspond to a predefined Fth value). ROC curves are useful tools but only allow addressing economic problems presenting cost loss ratios close to 1. Otherwise, the strategy to adopt consists in finding the value of Fth that maximize a predetermined economic value. Fth will be harmonized with the variable Dth defined in 4.1.4

-Par. 4.1.2 (Transmission strategies): the deterministic scenario is not described.

The first scenario is deterministic: the real attenuation is known and the MCS are chosen in real time as a function of the current propagation conditions. This is referred as the ACM strategy. However, is it true that the deterministic forecasts used for the simulation of a PCM-D strategy (scenario n ° 4) are not described. The first member of the PEARP ensemble, called the control member is arbitrarily chosen as the deterministic forecast. Then, for the PCM-D strategy simulation, only the first member of the PEARP ensemble has been considered in the training process and in the test process. -How was computed the "Mean" capacity of the different scenarios?

Satellite operators usually set the target link availability to 99.9% of the total transmission time. The methodology proposed in part 2 is used to compute, every 3 hours, the attenuation threshold Ath that, within a probability of 99.9%, will not be exceeded. The estimation of these attenuations allows programming a plan of MCS for all the simulation period (2014-2015) that will ensure the required availability. Table 3 gives the achievable capacity as a function of the MCS applied. The mean capacity is obtained averaging the time series of capacity thus obtained. -Pag.17, lines 8-10: please add some details

The Fig. 9 shows the mean capacities offered by the ACM, CCM and PCM strategies. In order to assess the interest of using probabilistic forecasts over deterministic ones, the PCM strategy has also been tested by using the control PEARP member only. This

strategy is referred as the PCM-D strategy.

-It is not clear the meaning of Dth: in line 32 (pag.18) is defined as a threshold on the forecast probability but in Line 1 (pag.19) it seems to be a threshold on the attenuation. Pag.19, line 2: is it Ath or Dth?

Thank you for pointing that out. Dth is the threshold of forecast probability from which the data storage must be preferred to the data transmission. Page 19, line 2, Ath is correct. However, page 19-line 1, Dth shall be replace by Ath.

- Pag.19, eq. (4): please add a reference for the equation (4) and clarify L (I guess it is total lost data over total transmitted data).

L is in fact the fraction of lost data over the total transmitted data.

- Fig.10: why is the y axis a "mean" value? Is it averaged over the 2 years (2014-2015) of simulations and Measurements?

In fact, the economic value is average over the period of reference of the simulation (2014-2015).

o Pag.20, line 4: please explain better.

The fig. 10 tells us that, for an attenuation threshold of 1.0 dB, the optimal decision threshold is 0.24 %. The ROC curves (fig. 8) indicates a decision threshold comprise between 0.7 and 1.4 %. This difference of appreciation is because the cost of misdetection is largely superior to the one of false alarm. The ROC curves do not integrate this information.

---

## Author Response (AR1)

Dear editor,

Please find below a detailed answer to reviewers' comments on manuscript.

We would like to thanks warmly the reviewers for their time and their help improving the manuscript.
* * *
**Reviewer 1**

NHESS journal is dedicated to studies on natural hazards and their consequences. Burton and all. define in [1] that "the term natural hazard refers to all atmospheric, hydrologic […] and wildfire phenomena that, because of their location, severity, and frequency, have the potential to affect humans, their structures, or their activities adversely". Satellite transmissions are today a must for the proper running of our societies but are vulnerable to rain falls. This paper proposes a strategy to reduce the costs involved by the atmospheric dynamic in the field of satellite communications. In that sense, it belong to the scope of NHESS journal. In [2] is an article published by NHESS. The latter aims at assessing the economic impacts of drought on sugar industry in China from the perspective of profit loss rate. This highlights that it could make sense to publish to NHESS journal an article dealing with the assessment of a weather dependent activity.

**[1] Burton, Ian, Kates, Robert W. (Robert William) and White, Gilbert F. (Gilbert Fowler), 1911-The environment as hazard. Oxford University Press, New York, 1978.**

**[2] TY - JOUR A1 - Wang, Y. A1 - Lin, L. A1 - Chen, H. T1 - Assessing the economic impacts of drought from the perspective of profit loss rate: a case study of the sugar industry in China JO - Nat. Hazards Earth Syst. Sci. J1 - NHESS VL - 15 IS - 7 SP - 1603 EP - 1616 Y1 - 2015/07/23 PB - Copernicus Publications SN - 1684-9981 UR - https://www.nat-hazards-earth-syst-sci.net/15/1603/2015/**
* * *
**Reviewer 2**

1-  **Part 1 :**

a.  **Pag.2 (Introduction): note that the ITU-R recommendations that are cited must be updated with ITU-R 618-13, 2017 and 837-7, 2017.**

References to ITU-R recommendations have been updated.

b.  **As concerning the recommendation ITU-R 618, which is further cited and adopted in the work, please check that the results are in line with the last recommendation 618-13, 2017 (e.g., the CCM results presented in section 4.1.3).**

The methodology being generic, there is no impact on the results.

**2- Part 2 :**

**a. Par. 2.1.1 (Beacon data): which is the precise frequency of Ka band signal of the Astra 3B satellite?**

The precise frequency of the Ka band beacon of the Astra 3B satellite is 20 GHZ. This information has been added to the manuscript (page 4, line 29)

**b. Par. 2.1.2 (Weather forecast model), please add some additional details concerning the adopted PEARP system:**
**o Please specify which is the total period over which the PEARP model is run: it should be over the years 2014 and 2015 (where the beacon measurements are available) but this detail should be explicated in this paragraph.**
**to give a complete presentation of the set-up of the adopted weather forecast model.**
**o The lead time of the weather forecast is not specified: since the ensemble forecast F includes 70 members per day (35 computed at 06.00 UTC and 35 computed at 18.00 UTC), I guess that the we are dealing with daily weather forecast (i.e., 70 members per each day of the forecasted period) but it should be better clarified.**
**o It is not clear if the PEARP members are time-series, over a certain period, of the rain accumulated every 3 hours: please, clarify this point.**
**o Please write explicitly in this paragraph that the time resolution of the weather forecast is 3 hours (which is the availability of the forecasted cumulated rain).**
**o Line 11, "Both lagged runs are used together": please clarify this sentence.**

Thanks for the suggestions. The 2.1.2 paragraph has been reworded to clarify the next elements:
Available archived data consists in three hours cumulative rain rate forecasts. The daily PEARP forecasts archived in 2014 and 2015 have been used on a 0.5°×0.5° grid in this study to build predictive attenuation distributions. Specifically, we use together 36h forecast run at 15 1800UTC day D with 24h forecasts run at 0600 UTC D+1. The 35 members of both runs, respectively 06.00 UTC and 18.00 UTC, have been used indiscriminately. In the manuscript, the ensemble forecast is denoted as F such as F=[member 1, member 2 ... member 70] where each member constitutes an estimation of the rain amount expressed in mm/3 hours.

**c. Par. 2.2 (The attenuation statistical prediction model):**

**o Pag.7, line 2: it should be called "complementary cumulative distribution of attenuation" instead of "inverse cumulative distribution of attenuation".**

Yes corrected.

**o How is it computed the complementary cumulative distribution of attenuation conditioned to the PEARP classes (the right probability of eq.1)? The probability that A>A\* (from the beacon measurements) is combined with the condition on PEARP classes but the procedure is not clear: measurements averaged over 5-minutes are compared with model outputs available every 3-hours.**

The process involved in the computation of the complementary cumulative distribution of attenuation conditioned to PEARP classes is now described (page 6).

**o Pag.7, line 8, "This methodology is equivalent to averaging the 70 rain attenuation distributions": this is true for a certain time horizon. It is not clear if the equation (1) is computed per each day of the simulated period (2014-2015).**
**o Concerning this section 2.2 and the applicability of the equation (1), some clarifications should be done. If I understood right, the available beacon measurements, combined with the forecasts (computed within the same period of measurements), are used to compute the probability P(A>A\*|F ⬚ ci). Once this probability is computed, it is stored such a "library" available in the operative context. When the satellite communication must be designed, a new forecast is produced for the target satellite-to-Earth transmission period. This forecast is used to compute P(F ⬚ ci) that, together with the probability in the library, allows the computation of the total probability in eq. (1). Please add this details in the paper.**

Again, thanks for the suggestions. The 2.2 paragraph was reworded to make theses clarifications.

**3- Part 3 :**

**a. Part 3.1 (Model reliability assessment) :**

o Pag. 8 line 23, "Time series of P(A>Ath|F) are computed": please clarify how are obtained the time series of the probability (and how are computed the curves of fig. 5). Once fixed Ath it comes that P(A>Ath|F), for a given F, is a number.

In the following, a positive event will be defined as the overrun of an attenuation threshold ATh alternatively set to 1 dB, 3 dB and 6 dB. The model developed provides the probability P(A > ATh F) which is the predictive probability that a positive event occurs as a function of the PEARP forecasts. Once fixed Ath it comes that P(A>Ath|F), for a given F, is a number. From the time series of the PEARP forecasts archived in 2014 and 2015, the time series of P(A>Ath|F) have been computed for the three value of ATh. Times series of attenuations observations and predictive probabilities P(A>Ath|F) allowed to trace the fig. 5.

o Fig.5: why is the climatological probability represented by a horizontal and a vertical line?

The reliability diagram shows the observed frequencies of an event as a function of its forecast probability. The diagonal line indicates the perfect reliability. It is conventional to represent the climatological probabilities of the event in the forecasts by the vertical line and in the observations by the horizontal line. The corrections have been made (at page 9, line 11-15 and in the legend of fig. 5).

o Fig.6: please clarify which is the rationale of the rank diagram.

In the case of an ensemble prediction system, the rank histogram of the position of the verifying observation with respect to the predicted ensemble values provides a measure of reliability (Talagrand et al. 1999). When the ensemble is reliable, say the verification and the evaluated system are from the same distribution the diagram is flat. If the diagram follows a U-like shape then the observation is often outside the ensemble suggesting the ensemble is under dispersive…

Theses explanations has been added in the manuscript (page 10 from line 7).

**b. Part 3.2 (Resolution and sharpness) :**

o All the symbols (TP, FN, FP, TN, FA) must be introduced and defined before being used.
o Fig.7: please check the block diagram, I guess that the (TN) and (FN) boxes should be inverted. Please check the consistency between the symbols used in diagram and the ones used in the equations (2) and (3): TP should be used instead of TD.

Thanks. Expected corrections have been made.

o Pag.12: Please give a definition of Fth and explain how is it chosen.3

FTh is defined as the decision threshold from which the predictive probability of exceeding a given attenuation threshold is considered significant enough for establishing protectives measures (for example to reduce the link capacity). In other words, FTh is the probability threshold from which the forecast of exceedance of a given attenuation threshold is considered as positive. This point has been clarified in the manuscript.

o Pag.13, lines 8-9: please clarify the sentence.

Fth have been replace by Dth in order to harmonize notations with part 4.1.4.
The following sentences were reworded: "Without any cost-loss ratio the potential value of the probabilistic binary forecast can be derived from the left and uppermost point of the curve. Actually, the cost-loss ratio is here non trivial and will be discussed in the section to come."

In the simplest case, it is the mean when the costs of False Alarms are in the same order of magnitude of the non-detection's one, Fth can be deduce from the ROC curves. The optimal Fth is then the one given by the left and uppermost point of the curve (each point of the ROC curve corresponds to a predefined Fth value). ROC curves are useful tools but only allow addressing economic problems presenting cost loss ratios close to 1. Otherwise, the strategy to adopt consists in finding the value of Fth that maximizes a predetermined economic value.

**4- Part 4 :**

**a. Part 4.1.2 (Transmission strategies): the deterministic scenario is not described.**

Thanks for pointing out this inconsistency. The first scenario is deterministic: the real attenuation is known and the MCS are chosen in real time as a function of the current propagation conditions. This is referred as the ACM strategy. However, is it true that the deterministic forecasts used for the simulation of a PCM-D strategy (scenario n ° 4) was not described. In the manuscript, the first member of the PEARP ensemble, called the control member is arbitrarily chosen as the deterministic forecast. This appears now in the manuscript (page 17, line 24-27).

**b. Table 4: some details should be given regarding the decision time, especially for the scenarios 3 and 4.**

The fifth line of tab. 4 (decision time) has been detailed. In case of ACM strategy, the MCS are chosen in near real time. In case of CCM strategy, the plan of MCS is programmed once, during the design phase of the satellite. In case of PCM or PCM-D strategy, the MCS are chosen one or two days in advance depending on the forecast availability.

**c. Part 4.1.3 (Results):**

**o How was computed the "Mean" capacity of the different scenarios?**

To evaluate the model performances, we have used the cross validation techniques. The available samples were partitioning in 10 subsets, 6 constituting the training dataset, 4 constituting the testing set. The procedure has been repeated 100 times and the capacities obtained have been averaged. This explanation has been added in the manuscript (page 18, line 17).

**o Pag.17, lines 8-10: please add some details.**

Details have been added.

**d. Part 4.1.4 (Economic value of the forecast):**

**o It is not clear the meaning of Dth: in line 32 (pag.18) is defined as a threshold on the forecast probability but in line 1 (pag.19) it seems to be a threshold on the attenuation. Pag.19, line 2: is it Ath or Dth?**

Thank you for pointing that out. Dth is the threshold of forecast probability from which the data storage must be preferred to the data transmission. Page 19, line 2, Ath is correct. However, page 19-line 1, Dth shall be replaced by Ath. This has been corrected.

**o Pag.19, eq. (4): please add a reference for the equation (4) and clarify L (I guess it is total lost data over total transmitted data).**

L is the ratio of total lost data over total transmitted data. Corrections and references have been added. (Page 20, line 31).

**o Fig.10: why is the y axis a "mean" value? Is it averaged over the 2 years (2014-2015) of simulations and measurements?**

Yes, the mean economic have been obtained averaging the economic values computed following the 4 for a simulation period of two years (2014-2015). This has been added in the manuscript.

**o Pag.20, line 4: please explain better.**

The strategy of maximization of the economic value leads, for an attenuation threshold of 1 dB, to an optimal DTh of 0.24 %. For a same attenuation threshold, the ROC curves in Fig.8 indicate a decision threshold comprised between 0.7 and 1.4 %. This difference of appreciation is because the cost of misdetection is larger than the one of false alarm. The ROC curves do not integrate this information. This explanation has been added (page 21, line 8).

**Technical corrections:**

 **Pag.2 (Introduction): please add a reference for the frequency band division in Table I.** A reference has been added.
 **Pag.4, par.2.1.1, lines 25-27: the sentences are not very clear, please rephrase.** The sentence has been reworded.

 **Pag.8, line 8: the acronym ROC is introduced here for the first time but it is not defined :** Yes, corrected

 **Pag.10, line 4: please check typo error :** Yes, corrected

 **Pag.11, line 11-12: the sentences is not clear, please rephrase.** The sentence has been reworded.

 **Pag.14, line 23: probably the symbol A\* should be replaced by A$_{Th}$ :** Yes, corrected

 **Pag.16, line 33: check typo errors and change table 3 with table 4 :** Yes, corrected

 **Pag.17, line 2: "transmission" instead of "transmitted":** Yes, corrected

 **Fig.9: please check typo error in the caption:** Yes, corrected

 **Pag.21, lines 1-2: please check typo errors :** Yes, corrected

[revised manuscript text omitted]